# Vertical Greening Systems: Technological Benefits, Progresses and Prospects

**Puyi Wang** [1] , **Yew Hoong Wong** [1,2,3,*] , **Chou Yong Tan** [1,2,3] , **Sheng Li** [1] and **Wen Tong Chong** [1,4,*]

1 Department of Mechanical Engineering, Faculty of Engineering, Universiti Malaya,
Kuala Lumpur 50603, Malaysia
2 Centre of Advanced Materials, Universiti Malaya, Kuala Lumpur 50603, Malaysia
3 Centre of Advanced Manufacturing and Material Processing, Universiti Malaya,
Kuala Lumpur 50603, Malaysia
4 Centre for Energy Sciences, Universiti Malaya, Kuala Lumpur 50603, Malaysia
* Correspondence: yhwong@um.edu.my (Y.H.W.); chong_wentong@um.edu.my (W.T.C)

**Abstract:** A vertical greening system is becoming increasingly crucial in resolving the energy crisis and environmental problems in a sustainable ecosystem. Researchers have conducted a comprehensive study on vertical greening systems from technology, functional and architectural perspectives. These include ecological, economic and social functions. Most of the current studies emphasize the benefits of vertical greening systems to the environment, while vertical greening technology and its socio-economic benefits receive insufficient attention. In order to study the vertical greening field in depth, this paper comprehensively and systematically summarizes vertical greening technology and functions. Meanwhile, based on the Web of Science (WOS), CiteSpace was used to analyze the relevant literature in the vertical greening field from 2012 to 2022, to explore the hot spots, development status and future trends of vertical greening technology, and to build a knowledge map in the vertical greening field. The research shows that as a low impact development technology, the vertical greening system has received the most extensive attention in the past few years. Air quality, microclimate regulation and energy have always been the focus and hot issues of people's attention. The future research directions are cooling effect, active system and indoor space. This study is aimed at promoting the future development of vertical greening system technology and providing reference and direction for researchers, planners and developers, as well as individuals interested in future urban and rural planning.

**Keywords:** vertical greening system; sustainable ecology; ecological benefit; low impact development technology



## 1. Introduction

Industrialization and urbanization have promoted the progress of modern civilization. As the city continues to develop and expand to its surrounding areas, the countryside gradually turns into the city. According to a report from United Nations (UN), the urbanization rate in developed countries is estimated to reach 83% by 2030 [1]. Urbanization has become the general trend of development all over the world. Sustained urbanization can lead to economic growth and social improvement. However, this rapid development leads to environmental degradation and climate change and brings huge pressure on the existing urban infrastructure [2]. Some examples of this include air pollution, rainwater runoff, severe urban heat island effect and biodiversity decline [3–6]. It also has adverse effects on human and socio-economic development, such as increased physical discomfort and health problems and increased demand for building cooling, resulting in increased energy consumption [6–9]. Therefore, sustainable measures need to be implemented and incorporated into new and existing development to mitigate the harmful effects of urbanization [6,10].

The demand for various resources, including building developments, land, water and energy, has dramatically increased due to global urbanization. Depending on information from the United Nations Environment Programme, buildings' construction and maintenance account for around 40% of the world's demand for primary energy and 33% of its environmental pollution [11]. Therefore, in this case, the implementation of sustainable development methods and plans to transform building components can achieve low-energy buildings [12] and save energy [13]. The burden on the urban environment and climate has been reduced by introducing sustainable green measures and technology [14]. Low impact development (LIDS), such as sustainable green buildings, design and practice techniques, has become the preferred choice for urban construction and planning in recent years [15]. According to the European Research and Innovation Policy Agenda, natural solutions (NBS) are key technologies to improve the sustainability of urban areas [16]. Greening System is the most popular sustainable building form, such as roof greening, wall greening and movable wall, which are often used as an aesthetic feature of architecture. At present, vertical greening technology can not only improve building performance [17] but also be an important measure to beautify the city and achieve sustainable development [18–23].

In urban areas, vertical greening does not occupy urban space [24]. In fact, vertical greening uses vegetation to cover buildings, which can bring environmental, economic, social and health benefits. It is an important feature of architectural design for sustainable development [25]. In terms of environment, the urban environment can be enhanced by promoting urban biodiversity [15,26], handling stormwater [27], air quality [28–31], and mitigating the Urban Heat Island (HUI) [32–34]. In addition, a green wall is an alternative form of constructed textiles, which has been well recognized and widely applied worldwide for various wastewater treatments, particularly in green water treatment [16,35]. In terms of social health, it provides aesthetic and therapeutic benefits, improves the city image [21,36,37], reduces noise [38,39], improves the lifestyle of residents, increases architectural value [40] and complements thermal [41] and acoustic protection [42,43]. Economically, cooling can save energy, reduce energy consumption [25,44,45] and prolong the service life of the roof membrane [35,46,47]. The wall greening has greater potential than the roof greening. Because of the limitation of municipal space, the greening degree of the external wall can be twice as much as that of the roof and the ground [24,26,27]. Vegetation can regulate the microclimate climate in winter [28] and summer [29], providing heat preservation and shading [30,32,33] and evaporative cooling effects [34]. Vegetation absorbs a large amount of solar radiation in summer [36], which can reduce the temperature [39] and increase the humidity under the transpiration so that the wall surface temperature is lower than the surrounding temperature [37,39–41]. Recent research shows that green systems can improve the thermal performance of buildings [44] and regulate the capacity of heat gain and loss through the basic mechanism of thermal fluid and energy conversion, which enhances indoor thermal comfort, lowers the energy required for building heating or cooling, and results in energy savings [17,42–44,46]. Altogether, green walls are a fit-all solution to reduce building energy demand and mitigate UHI and air pollution, providing healthy living conditions. It is the solution to urban environmental sustainability [31,44,47].

As scholars pay more and more attention to all aspects of the vertical greening system, they have conducted research and published papers. So far, some vertical greening system technologies have been proposed, which can help researchers to keep pace with the development of knowledge in this field. This could make it more challenging for researchers to understand their area of study and the current context. It should be noted that the development of knowledge is a dynamic process that involves constant adjustment, enhancement and renewal. Although the vertical greening system has developed rapidly in the industry and research field, there are still many problems to be further improved and strengthened in achieving sustainable development and the optimization and promotion of technology. Therefore, this study systematically analyzes the vertical greening technology and its environmental, economic and social functions by combing the existing literature and helps us to visually and quantitatively analyze the keywords, literature co-occurrence and

time zone map in the vertical greening field in combination with CiteSpace software. Help us better discover the existing vertical greening technology, functions, research progress, hot spots and future research directions and trends so as to emphasize the direction for future research in this field.

## 2. Vertical Greening System

A vertical greening system refers to all systems that can green vertical surfaces. It is related to the selection of plant species, including all solutions designed to plant plants on the wall or in the building [48]. The vertical greening design shall consider the climatic conditions, the structural performance of buildings (structures), morphological layout (orientation, height), plant characteristics, construction and maintenance costs, safety and durability and other factors. Figure 1 is a classification of vertical greening technologies.

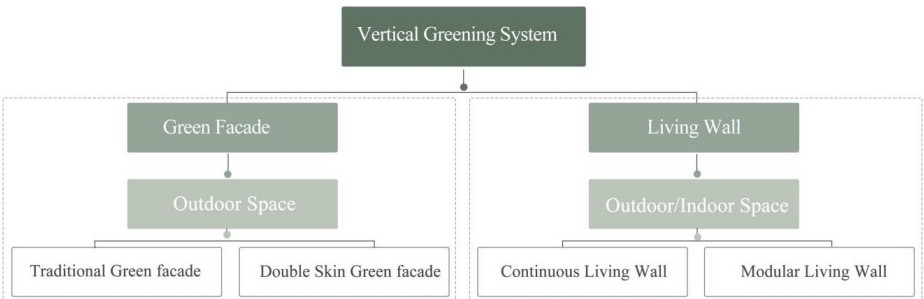

**Figure 1.** Classification of vertical greening system.

### 2.1. Vertical Greening Technology

In Figure 2, vertical greening technologies are classified as green facades [49] and living walls [50] based on varied vegetation and substrates [51,52]. The former usually refers to climbing plants growing along the wall, while the latter includes materials and technologies supporting more kinds of plants to form uniform growth along the wall. The arrangement is referred to as a green facade when the growth medium is the floor. There are two different kinds: a direct facade, such as a traditional facade, and an indirect facade, such as continuous guidance, modular lattice or double skin facade. The system is known as a living wall or green wall when the growing media is integrated into the building wall [53]. There are two varieties of green walls: continuous green walls that use a continuous panel or geomembrane and flexible green walls that use a tray, container, soft bag or flowerpot. According to the degree of difficulty, the green wall is divided into extensive or dense systems. The green facade is categorized as an extensive system, which is simple to construct and requires teensy future maintenance, while the living wall is classified as an intensive system, which is more complicated in plan and has a higher degree of maintenance [50]. Table 1 lists different vertical greening technologies.

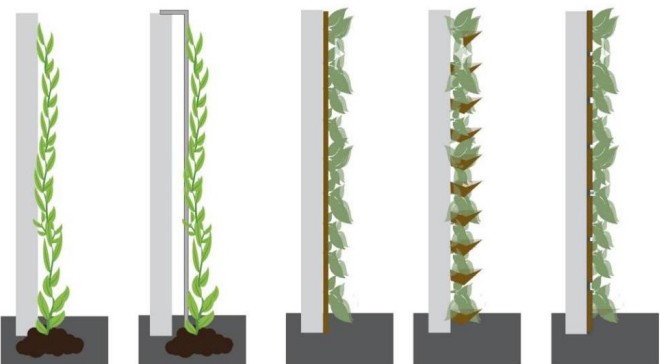

**Figure 2.** Vertical greening technology.

**Table 1.** Existing vertical greening technology.

| Type | Climbing | Traction | Container | Basin | Cloth Bag | Plate Slot | Modular |
|---|---|---|---|---|---|---|---|
| Illustration | | | | | | | |
| Case | | | | | | | |
| Cost | Low | Medium | High | Medium | Low | Low | High |
| Plant | Single | | | Multiple | | Fit | Multiple |
| Effect | Medium | | Fine | | Medium | | Fine |
| Maintain | Inconvenient | | Convenient | | Inconvenient | Convenient | |
| Weight (Kg/m$^2$) | 2.5 | 30–60 | 50–100 | | 25–50 | 60–120 | 25–50 |
| Tickness (mm) | 100–200 | | 300–500 | | 250–400 | 300–500 | 200–500 |
| Substrate | Unlimited | | Less limit | | Limit | | Less limit |
| Time | Slow | | Medium | | Faster | | |

## 2.2. Components of Vertical Greening

The Figure 3 vertical greening system is mainly composed of vegetation, substrate, supporting structure, irrigation and drainage system, thus forming a complete green ecosystem.

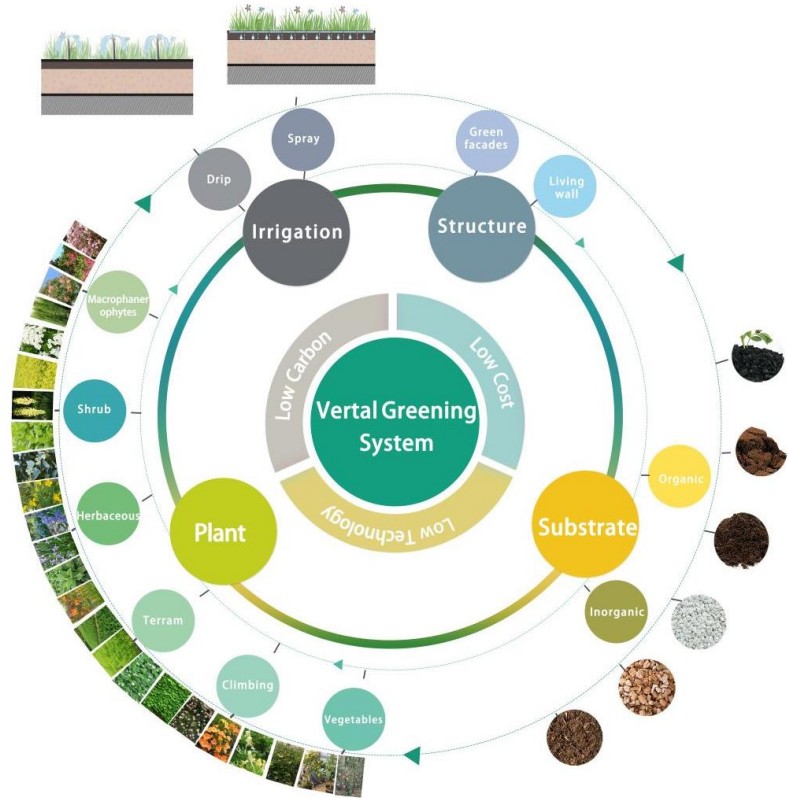

**Figure 3.** Composition of vertical greening system.

### 2.2.1. Plants

Plants are an important component of vertical greening systems. They not only provide ecosystem services but also undertake most of the environmental benefits and urban beautification of the vertical greening system [54]. The green wall [35] is made up of climbing plants. At the same time, the modular green wall is manufactured of non-climbing potted plants.

(1)  Climbing plants

Climbing plants are widely used in vertical greening and are economical and practical plants. It can be used for the greening of vertical sides of various walls, retaining walls, bridges, buildings, etc. Climbing plants can be attached to any vertical surface. In some countries, it is common to plant plants on the outer walls of small buildings [53]. It takes about 3–5 years for some species to reach a height of 5–25 meters to achieve full plant coverage [50,55]. In one study, leaf densities of several climbing plants (*Hereda helix, Lonicera japonica) and deciduous plants (Parthenocissus quinquefoliate, Clematis SP*) were compared after one year. The results showed that Parthenocissus quinquefoliate provided a higher leaf density, and all species were met with muted green [56]. Vertical greening systems generally utilize Parthenocissus and Hydrangea, as well as the evergreen specialty spear and ivy [35]. In addition, winding climbing plants that need endorses, such as metal frames and wires, include Humulus lupulus and Capsis radians. The trailing shrub is usually used in the grid structure, including Forsythia suspense and Pyracantha atalantiodes.

(2)  Shrubs and succulents plants

The new living wall makes vertical greening unrestricted, not limited, allowing more plant species, such as the integration of shrubs, grass and some plants, and some potted plants, such as asparagus, Begonia and Dragon blood tree. Hedera helix and Parthenocissus are the most frequently used plant species [49].

Recently, the modular living wall uses succulent plants instead of perennial plants and shrubs. Drought-tolerant plant species [17] can help to reduce irrigation demand. The maintenance cost of these plant varieties is also quite low, contributing to the system's weight reduction. However, the surface of succulent plants is flat and suitable for small walls. On the larger wall surface, the use of perennial plants and shrubs can generate more decorative landscapes because these plants are diverse in color, shape and texture. Studies in Japan have shown that [57] shrubs can be supported on inclined surfaces.

(3)  Hydroponic plants

The hydroponic system enables more kinds of plants to grow in different developmental states [58]. In these cases, the vegetation type is chosen according to the required scene, climate and aesthetic effect [59,60], and the appropriate irrigation and nutrition are assisted in achieving the healthy growth and development of plants. Therefore, plant composition: Color, flowering, leaves and morphology are important to plant growth [52]. However, under the concept of sustainable development, vegetation must have low irrigation demand, sufficient water and nutrients for plant growth (such as using local plants), and adapt to local exposure conditions (such as sunshine, semi shading or shading) and the best climate conditions (such as wind, rain, high temperature, drought and frost).

### 2.2.2. Irrigation

The selection of irrigation type is based on the system type, the plant used (style, size, growth period, root system) and seasonal changes [52]. Modular Green exterior walls and living walls require irrigation systems to provide necessary water irrigation for plant growth. Furthermore, the growth of vegetation can be aided by nutrients, fertilizers, minerals, phosphates, micronutrients or hydroponics substances [52]. The continuous living wall passes through the upper part of the structure and connects to the central irrigation system. For the uninterrupted living wall, the irrigation system makes the water and nutrients evenly distributed along the surface layer. Some living walls in the form of

pallets are supported on the top surface of the modules for inserting grooves for irrigation pipes. The tray includes holes on the grooves for pouring the growth medium under the action of gravity [60–64]. In order to allow extra water to irrigate the modules below, the tray's bottom has drainage holes [52]. Irrigation pipes and joints can be made of a variety of materials (such as rubber, plastic, thermoplastic pipes, silica gel and irrigation pipes), including different irrigation methods (such as drip irrigation, sprinkler irrigation, hole irrigation and pipe irrigation), as well as their distribution and strength, which can meet the irrigation needs of plants.

Some living walls also mentioned approaches for reducing the usage of treated water, such as storing water from building roofs [60], reusing the versus water in the drainage system [65], monitoring the stormwater management demand [60] and controlling the water level, irrigation time and extreme weather conditions by installing sensors [66,67], in order to minimize the wastage. Other than the living wall, whether modular [68,69] or continuous [66,67], it also refers to the installation of a ditch on the basis of the system, the recovery of excess water for storage and re-introduction into the irrigation system. Another strategy is exploited by sensors to quantify nutrient requirements in growth media. This is extremely important to minimize the consumption of nutrients and meet the needs of plants.

### 2.2.3. Substrate

The substrate is an important aspect of vertical greening systems. In some cases, the choice of substrate will affect the function of the green wall system. Existing studies indicate that the matrix can effectively remove pollutants. Similarly, the efficiency of green wall systems in wastewater purification and pollutant removal is largely impacted by the efficiency and selection of matrix materials [35]. In order to ensure the effective operation of the green wall, appropriate materials are required as the base materials [70]. Recent research has shown that using biochar [71] and alum sludge [72] as materials improves the ability to remove pollutants while also promoting plant growth.

The matrix in the green wall is made up of both organic and inorganic substances [57,64,73], or it has an added element of the inorganic matrix, typically foam, to lessen the weight. The matrix is a crucial component that influences the wall's steady state and structural load-carrying capacity [35,74]. The physical characteristics of these matrices are used in the majority of green wall designs to achieve porous structure, surface area, uptake capacity and water-holding capacity [35,60,65]. The matrix can be enhanced by using nutrients in plant growth, such as blends of organic and inorganic fertilizers, metal compounds, minerals, plant nutrition, hormones and other additions [64]. According to some modular living wall systems, the geotextile bag is put with the growing medium inside to keep it from going out. These bags can completely bridge the module, granting the insertion of multiple plants [60], or they can bridge the growth media of each plant separately [58,61]. Every plant also requires a fixed front cover to keep the growth medium from slipping off [64].

### 2.2.4. Structure

Conventional or direct green external walls typically have no supporting framework. They rely on climbing plants' capacity to adhere to vertical surfaces. However, there is a higher risk of falling when the vegetation is too thick. The "double-layer external wall" forms a certain range of gap between the building surface and the plants as an indirect green exterior wall. The use of support structures prevents vegetation from collapsing. Whether continuous or modular, it can stabilize and sustain the weight of the foliage and increase the system's resilience to harsh weather influences such as wind, rain and snow.

Most indirect green external wall supporting structures consist of continuous or modular guides [52,63,75]. Climbing plants with thick leaves can be fixed and supported by a steel structure and stretch cable. Smaller spacing between the grid and the steel wire mesh allows for supporting plants with sluggish growth rates [55]. Some indirect green

curtain-wall solutions, primarily modular flower racks, have independent supporting structures and pots with substrates inside, which enable pieces to be suspended along the wall at various heights. The modular green wall adopts an arc-shaped grid, which gives the wall a sense of rhythm and three-dimensional feeling [62,75]. A frame is generally included in a living wall to support the components and plants. Through a frame mounted on the wall, the movable wall creates a vacancy between the system and the surface. This frame supports the floor and protects the walls from moisture. The next layer is supported by the bottom and fastened to it with nails. It is shielded from the base by a permeable, flexible and root-resistant shielding layer. After that, the outer mesh layer was installed in cloth bags [66,67] for plant planting.

Modular living walls can take many forms (trays, containers, flowerpot bricks or flexible bags) and require distinct structures. Modular trays are usually composed of multiple interlocking parts and are made of soft materials, such as plastic or metal plates [60–64,73], in order to ensure the continuity of the system. Each component usually includes a side interlocking system for interconnection. These module elements may also include front cover trays and containers forming a grid to prevent plants from falling, which are generally fixed on vertical and/or horizontal frames connected to the surface. The rear surface may include hooks or mounting brackets [52,60,64] for hanging on a frame connected to the perpendicular surface. Multiple plants can be installed in every component of the same sequence using modular containers. They are usually constructed of polymeric materials and, due to their shape and structure, have a significant visual impact on the building exterior.

### 3. Bibliometric Analysis of Vertical Greening System Research

Scientometrics is a branch of informatics. It quantitatively analyzes the patterns in the scientific literature to find out the latest trends and knowledge structures in the research field [76]. The bibliographic analysis is an approach for quantitative analysis of articles, mainly for objective and systematic analysis of published papers. In 1969, this method was first commonly used in the literature to review evolution and predict upcoming trends [77]. Chen et al. analyzed the field of regenerative medicine using CiteSpace and found the latest development trend in this field [78]. Li et al. utilized by CiteSpace to examine the construction field and build a knowledge map of the building information model [79]. CiteSpace has been widely used in various fields and has achieved good results. However, in the area of vertical greening, there are still few bibliometric studies. This section employs bibliometrics to quantitatively analyze the vertical greening system field, primarily introducing the number of publications, research hotspots and citation evolution in significant countries in this field.

### 3.1. Methods and Tools

In order to obtain reliable and comprehensive literature data, the Web of Science (WOS) Core Collection Database is involved as the resource. The database is a text file, including the number of variables such as title, author, year of publication, language, abstract, keywords and references. Data collection flow is shown in Figure 4. The search time in WOS is "2012.01-2022.05", and the theme is "Green Wall", "Living Wall" or "Vertical Gardens" or "Vertical Green System" or "Biowalls" or "Green Vertical Systems" as the subject, and a total of 547 articles were obtained. To further ensure the rationality of the article, the search results were thoroughly checked, sorted out and some irrelevant sample data was deleted; the search scope was reduced manually. Firstly, irrelevant documents such as meetings and essay solicitations were deleted. Secondly, we deleted some non-academic literature and filtered out less representative record types. Our final sample included 406 original research articles and 43 review articles. Using this database, CiteSpace was used for further analysis and processing.

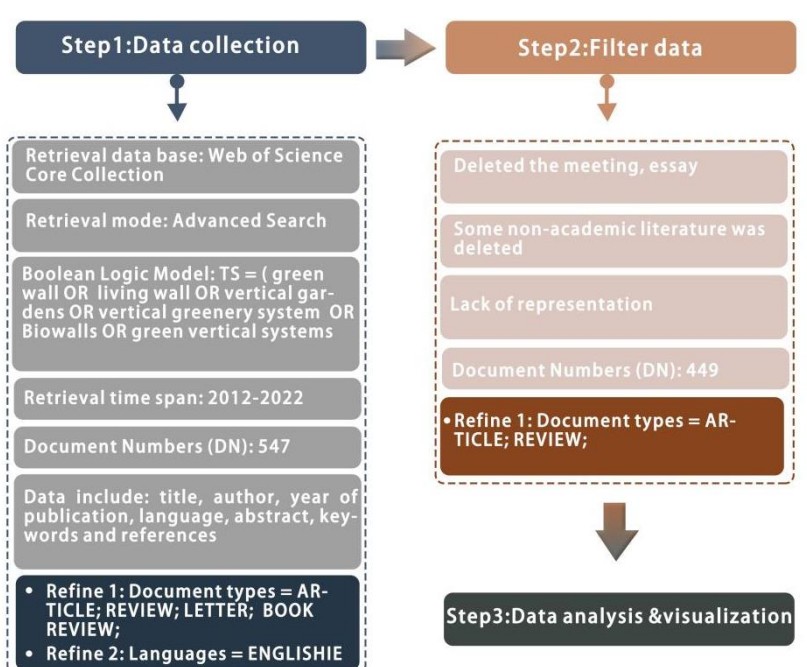

**Figure 4.** 2012–2022 Document data collection flow chart.

## 3.2. Research Status

It can be obviously observed in Figure 5 that the number of publications in the field of vertical greening is increasing exponentially. From 2012 to 2022, the number of documents increased year by year. Specifically, according to the change trajectory of the number of published research papers, three stages can be identified: the first stage (2012–2014) and the gradual rise stage; the second stage (2015–2019) is a stable development period; and the third stage (2010–2022) is a period of rapid progress. In 2021, the number of documents will reach a peak of 98. At this stage, the research results are relatively high and gradually mature. So far, the average number of documents released at this stage is 68, which shows that people pay greater attention to the research of vertical greening. At the same time, it can be predicted that in the near future, the number of papers related to vertical greening will increase rapidly, and new technologies and related strategies will appear.

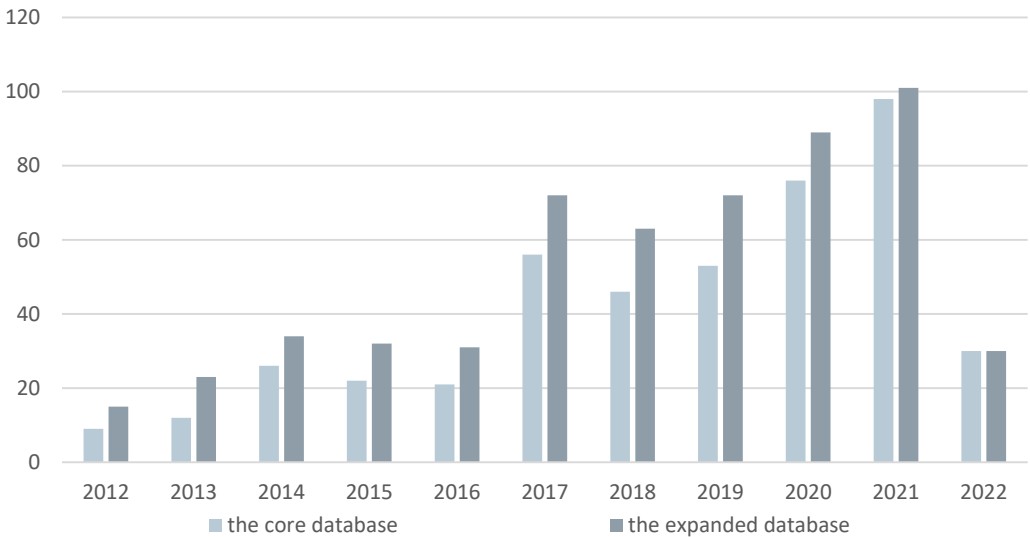

**Figure 5.** Number of documents issued for vertical greening in 2012–2022.

Figure 6 shows the hot keywords in the field of vertical greening systems in the past decade. The high-frequency keywords are "Urban heat island", "Thermal performance", "Energy performance" and "Climate change", which means that it has been the focus of vertical greening research for many years. "Thermal performance" is the support point of the whole network, which lays the foundation for the stability of the entire network and is the main research hotspot. The second level focuses on "Indoor air quality" and "Life cycle assessment".

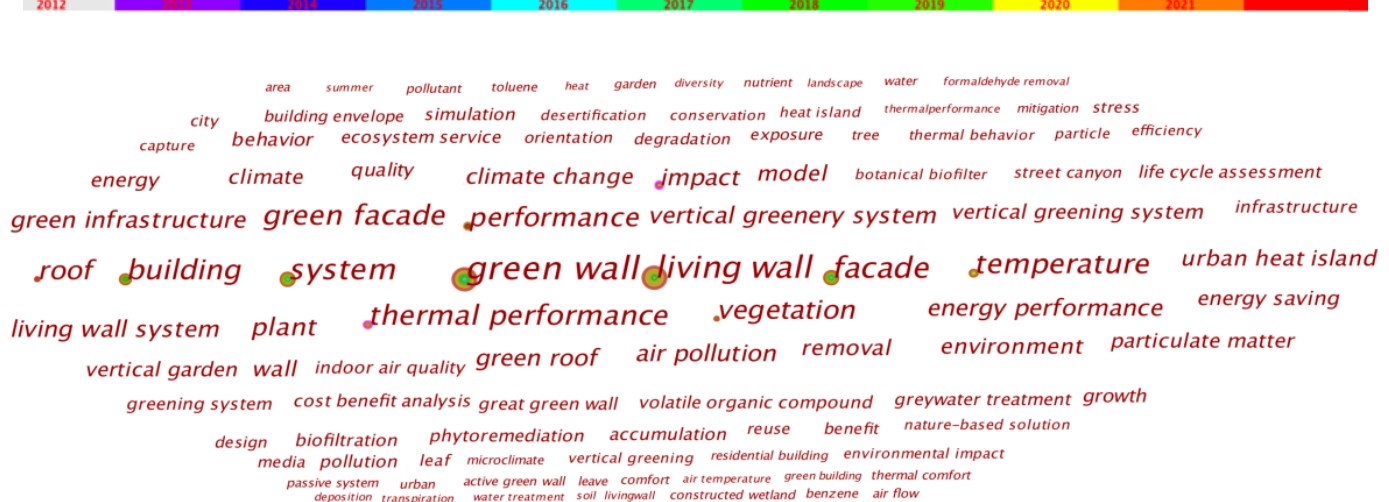

**Figure 6.** 2012–2022 High frequency keyword co-occurrence.

Figure 7 visually shows the evolution process of this field, which is useful for predicting the development trend in the next few years. As shown in the figure, air quality, climate, energy, refrigeration and thermal performance have been the eternal topics in the field of vertical greening in the past 10 years. It has mainly gone through three stages. The first stage (2012–2016) is the initial stage of the vertical greening system, and it began to expand the relevant knowledge and research in this field gradually. With the development of urbanization, the demand for urban heat islands and energy is increasing, resulting in a series of environmental and public health consequences. People gradually increase green spaces through vertical greening, roof greening and tree greening, which can effectively reduce the urban heat island effect and save energy. At the same time, green space also increases the cooling effect of water and wind. Therefore, vertical greening is an effective method to solve the urban air pollution level. In the second stage (2017–2019), with the rise and development of vertical greening technology, many countries have applied it to gray water treatment, indoor space removal of organic pollutants, inorganic pollutants, $CO_2$ and particulate matter, and pollutants in street canyons to improve air quality and achieve good benefits. At the same time, there is a series of international evidence that indoor plants have a direct and beneficial impact on human health, social and mental health and work efficiency. In the third stage (2010 to now), under the background of 2019 coronavirus disease, people have entered the era of user-centered mobile Internet. Vertical greening technology has been continuously improved, and some new technologies have been integrated while solving the problems of air quality, microclimate regulation and energy. We also need to pay more attention to the comfort of user groups.

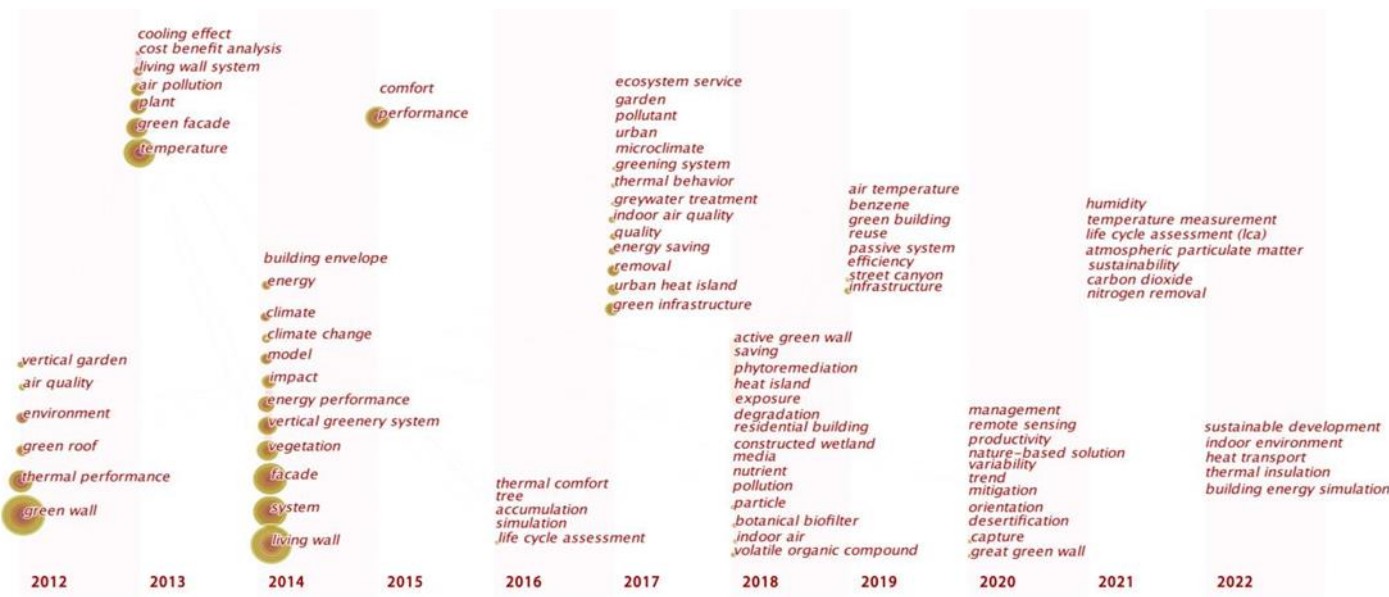

**Figure 7.** 2012–2022 Keyword co-occurrence time zone map.

Figure 8 shows the map of transnational cooperation. On the whole, a collaboration between countries/regions is relatively close. China has the closest cooperation with the United States and Australia. American countries often cooperate with Germany. In China, the research primarily examines and assesses the subtropical external space vertical greening system. Its energy-saving benefits through the simulation and measurement of the vertical greening system. In Australia and Italy, the movable wall system will be used to treat the wastewater from showers and washing basins. Innovative and beautiful living walls can not only be used for household-scale grey water treatment but also provide key comfort and microclimate benefits for our cities [57]. Moreover, there could be further improvements to the technology of removing VOC, $CO_2$ and PM in the indoor space. Plants and substrates in the functional green wall/modular plant biofiltration system can effectively remove indoor PM, especially ferns [59,60].

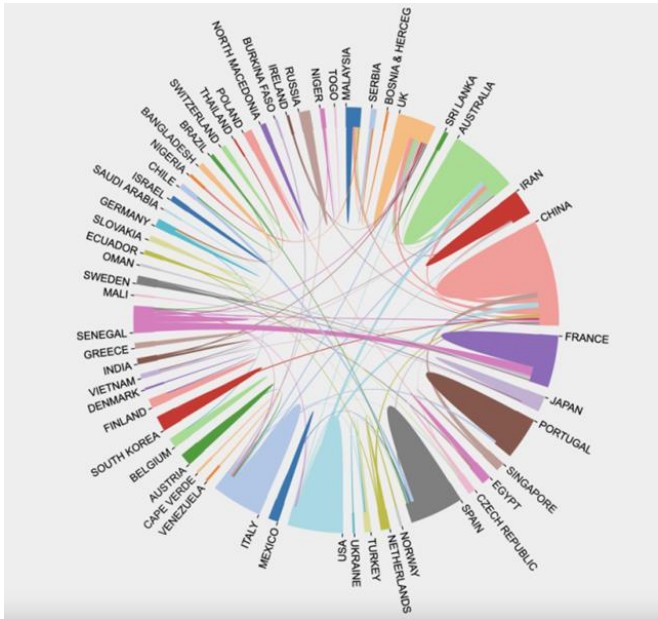

**Figure 8.** The cross-country collaborations visualization map in vertical greening.

Figure 9 shows the analysis results of document clustering and timeline from CiteSpace. The documents in the same cluster are highly homogeneous. In the cluster diagram, the warm color represents the research focus in this field in recent years, while the cold color represents the early time of research. The rise, prosperity and decline process of specific cluster research can be clearer with the timeline chart, showing the temporal characteristics of the field. Among them, large nodes or nodes with a red tree ring represent documents that are highly cited or have citation explosion. It can be included that the current distribution fields of vertical greening system research are mainly Clusters # 0, 1, 3, 4 and 6. Among them, # 0, 1 and 3 have highly concentrated nodes with citation bursts, and # 1 is still active; Clusters # 4 and 6 also seem to have the latest publications that cited citations, which are still active.

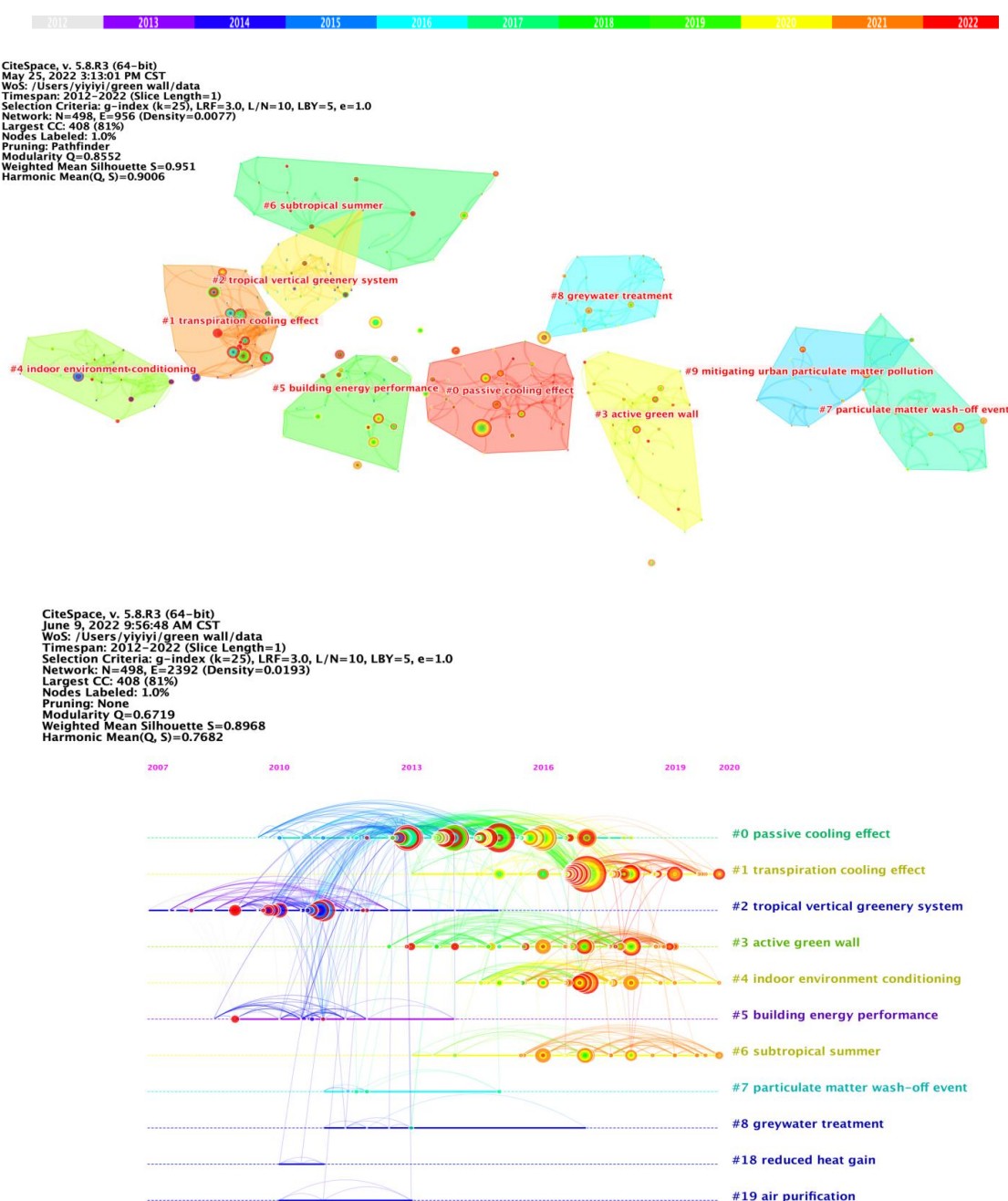

**Figure 9.** 2012–2022 Visualization of the reference co-location clustering network and timeline.

According to the literature analysis from 2012 to 2022, the research on vertical greening is mainly based on the research on green buildings. The research methods, seasons, plants and matrix types in this field are shown in Figure 10.

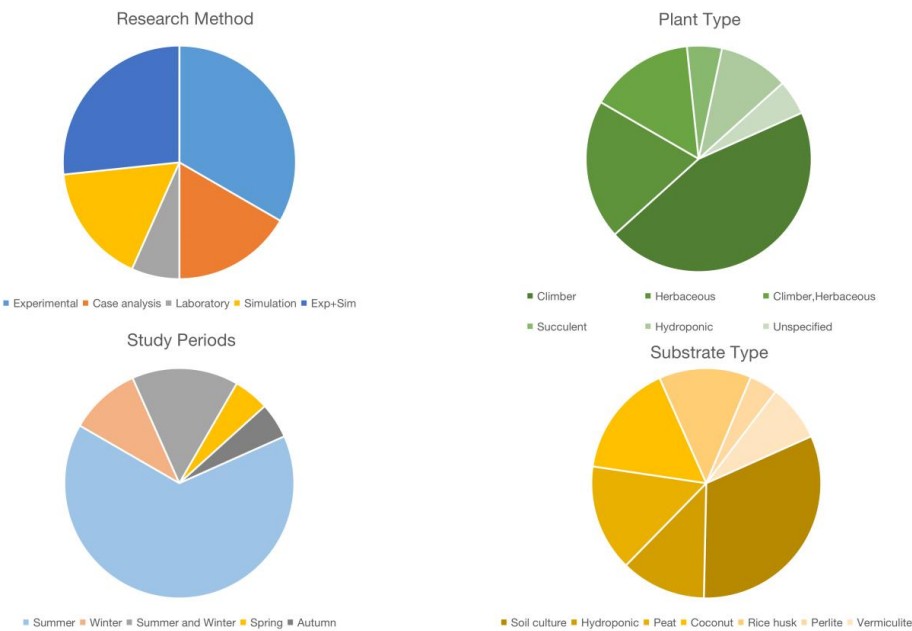

**Figure 10.** Review the details of previous studies.

## 4. Functions of Vertical Greening System

Figure 11 is a functional overview of the vertical greening system. The results show that the green wall in the urban environment brings benefits to users (Aesthetic and Psychological), buildings and the surrounding environment (Improving Air Quality, Reducing Heat Island Effect, and Treating Sewage), society (Mitigating Noise) and economy (Energy Conservation).

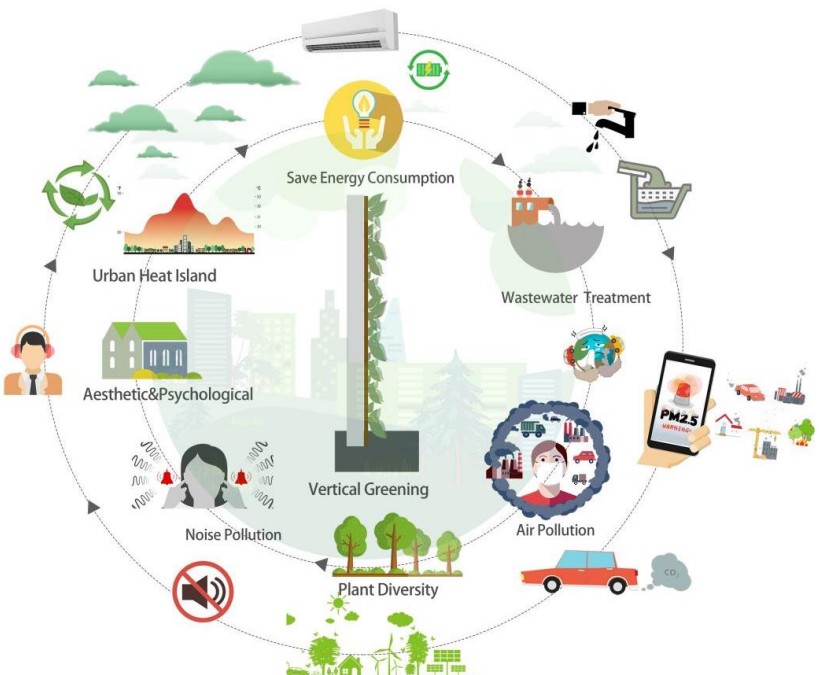

**Figure 11.** Summary of vertical greening system function.

### 4.1. Environmental Functions

### 4.1.1. Regulating Climate

The plants in vertical greening absorb the incident solar radiation through their physiological processes so as to carry out photosynthesis and transpiration and reduce the wall surface temperature [80]; building cooling is achieved through the evapotranspiration process, as the leaves of plants lose water and convert solar radiation into extra heat. This physical process produces what is called "evaporative cooling". Evaporative cooling of leaves depends on the plant species, and substrate humidity and temperature variations influence it. The impact of a dry environment or wind will increase the evapotranspiration of plants [81]. In addition, plants can also control and modify the wind direction and wind speed. The hot air is caused by convection and rises through the space between plants and buildings, forming a local breeze and alleviating the urban microenvironment. Table 2 is the relevant literature on the cooling effect of vertical greening.

**Table 2.** The literature on the cooling effect of vertical greening.

| Type | Country | Period | Plant | Reduction Temperature | Method |
|---|---|---|---|---|---|
| Green Facades | Germany [49] | Winter | Vines Boston Ivy | 3° | |
| | England [82] | One Year | Hedera Helix | Summer 1.7°–9.5° | |
| Direct Facade | Deflt [83] | | Hedera Helix | 1.2° | |
| Indirect Facade | Rotterdam [83] | Autumn | H-helix, Vitis, Clematis, Jasmine and Pyracantha | 2.7° | |
| Living Wall System | Benthuizen [83] | | No climbers | – | Experiment |
| Living Wall System | Singapore [84] | Summer | – | 6°–10° | |
| Direct Facade | China [85] | Summer | Parthenocissus Tricuspidata | 2.57°–4.67° | |
| Green Facades | UK [86] | Summer | Hedera Helix, Stachys Byzantina | 7° | |
| Living Wall System | UK [87] | Winter | Hedera Helix | 0.5° | |
| Green Facades | USA [88] | Summer | Ivy | 0.7°–12.6° | |
| | USA [89] | Summer | Parthenocissus Tricuspidata | 7.9° | Simulation |
| Double-skingreen Facade | Spain [81] | All Year | Wisteria Sinensis | 5.5°–17.62° | Experiment |
| Greenwall | China [90] | Summer | Pumila, Venusta, Corymbosa | 2°–5° | |
| Greenwall | Madrid [91] | | – | 4.5°–8.2° | Exp+Sim |
| Vertical Greenery Systems | Singapore [43] | Summer | Climber Plants | 4.36° | |
| Green Facades | Japan [92] | | Bitter Melon, Morning Glory, Sword Bean, Kudzu, Apios | 3.7°–11.3° | Experiment |
| | Slovenia [93] | | Phaseolus Vulgaris "Anellino Verde" | 4° | |
| | Spain [94] | | Parthenocissus Tricuspidata | 15–16.4 | |

### 4.1.2. Improvement of Air Quality

Air pollution will directly threaten human health and shorten life spans. With the development and progress of vertical greening technology, vertical greening systems can effectively remove particles in the air.

(1) Outdoor air quality

According to Table 3, green walls are more useful in removing PM than gas pollutants. By absorbing PM and lowering environmental concentration, plant leaves can improve the quality of outdoor air [57,72,73]. Green walls can provide practical pollutant collection through literature review and summary, enhancing air quality. However, plant species, size, wind direction, leaf area index and humidity all play a role in how much pollution is reduced by green walls. Green walls show better results in collecting fine and ultra-fine particles than coarse particles, and they are more effective at removing particulate matter than gas pollutants. The effects of green walls with different heights and different seasons

on decontamination are not significantly different, but the topographic differences of living walls affect the capture of pollutants.

**Table 3.** Literature on vertical greening to improve outdoor air quality.

| Region | Type | Plant | Conclusion | Method |
|---|---|---|---|---|
| Urban Toronto [95,96] | Green Roofs and Green Walls | Shrubs | The removal rate of PM10 is 1.37 mg/year. | Experiment |
| Road [97,98] | Vegetation Barriers | – | The reduction of pollutant concentration is due to the dispersion and deposition of green walls. | ENVI-met |
| New Street Railway Station [99] | Living Wall | Buxus Sempervirens L., Hebe Albicans Cockayne, Thymus Vulgaris L. and Hebe X Youngii | Living wall plants have considerable potential in removing particulate pollutants from the atmosphere. | |
| Road [100] | Living Wall | Twenty species of living wall plants | The average capture of PM1 particles by the living wall of 100 square centimeters is $122.08 \pm 6.9 \times 107$, PM2.5 particles $8.24 \pm 0.72 \times 107$, PM10 particles $4.45 \pm 0.33 \times 107$. | Experiment |
| Road [101] | Green Wall | Heuchera Villosa Michx, Helleborus × sternii Turrill, Bergenia cordifolia (Haw.) Sternb. and Hedera Helix L | The recovery of PM capture ability of four green wall species after rainfall was studied. Green wall has the potential to capture PM all year round. | |
| Road and Woodland [102] | Living Walls | Hedera Helix L | The number of particles collected on the front of leaves was more than that on the back, and there was no significant difference in height and season. | |
| Developing Countries [103] | Green Facades | Vernonia Elaeagnifolia | For ($SO_2$), the removal rates in dry and wet weather are $1.11 \times 10^{-6}$ s$^{-1}$ and $1.05 \times 10^{-6}$ s$^{-1}$ respectively | CFD |
| Road [104] | Green Wall | | At pedestrian height (1.4 m), green walls are an effective barrier to reduce exposure to pollutants and air quality deteriorates from 4 m. | ENVI-Met + Experiment |
| Street Canyons [29] | | | By planting vegetation in street canyons to increase sediment, street concentrations in these canyons can be reduced by 40% for $NO_2$ and 60% for PM. | CFD |

(2)　Indoor air quality

　　The vertical greening system gradually turns from outdoor to indoor space. Indoor green walls are divided into active and passive forms, which are mainly used to reduce indoor air pollutants [105,106]. The active system has a ventilator that can simultaneously deal with a lot of air pollutants at a low cost while forcing air through the matrix of the vertical greening and the plant rooting system [107,108]. In passive systems, polluted air is simply absorbed and removed by the green wall matrix and plant leaves. The advancement of vertical greening technology is currently moving toward the incorporation of ventilation systems and green walls into building air conditioning systems. Because they increase plant density, are arranged vertically, and allow for the effective passage of polluted air through the matrix, green walls are preferable to pot plants [109]. Recent studies have shown that active green walls have high phytoremediation capacity and can repair a variety of air pollutants. Table 4 shows the relevant literature on indoor vertical greening to remove indoor pollutants.

**Table 4.** The Literature on indoor vertical greening to remove indoor pollutants.

| Inorganic Pollutant | Plant | Conclusion |
|---|---|---|
| $NO_2$ [110] | Spathiphyllum Wallisii and Syngonium Podophyllum | Average $NO_2$ clean air delivery rate of 661.32 and 550.8 $m^3 \cdot h^{-1} \cdot m^{-3}$ of biofilter substrate for the respective plant species. |
| $CO_2$ [111] | – | The indoor plant wall of 5.72 $m^2$ can reduce the $CO_2$ concentration of 38.88 $m^3$ room from 2000 to 800 ppm in one hour. 1 $m^2$ of active green wall can significantly reduce indoor carbon dioxide. |
| PM [111,112] | Chlorophytum Comosum | The system recorded removal efficiencies were $53.35 \pm 9.73\%$ for total suspend particles, $53.51 \pm 15.99\%$ for PM10, and $48.21 \pm 14.71\%$ for PM2.5. |
| VOC [113] | Different plant species | The significant single removal rates (spres) of toluene and formaldehyde were 91.7% and 98.7% respectively. |

### 4.1.3. Sewage Treatment

Vertical greening systems have played a great potential in water savings in urban residential areas. They can not only relieve the pressure on urban sewage treatment plants but also provide water for gardens, green spaces, golf courses and toilets. By implementing these green walls, 40–50% of water can be saved [114].

The principle of vertical greening wastewater treatment has three aspects: physical mode (filtration and sedimentation), chemical mode (reaction and adsorption) and biological mode (removal of pollutants in wastewater through plants and microorganisms in the matrix) because water vertically penetrates down through the matrix. The substrate serves as a surface for bacteria to adhere to while filtering out organic substances. At the same time, they can also serve as the basis for supporting plants [70]. In addition to promoting oxygen transfer to the surrounding soil, which enables microbes to settle and disparage organic contaminants, vegetation offers a suitable environment for microorganisms that capture nutrients from wastewater and degrade organic contaminants [115]. The performance of the wastewater treatment green wall system is determined by major components such as plant selection, fluid load (flow/surface area ratio), water content inside the medium, ambient air quality and ash inflow [35,116].

### 4.2. Economical Function

The Heating Ventilation and Air Conditioning (HVAC) systems are primarily responsible for indoor thermal comfort and ventilation [117]. The development of the green building concept affects the performance of residents and leads to an increase in energy load. Using a vertical greening system on the building shell is a passive energy-saving technology in buildings and one of the solutions for lowering building energy consumption [118]. It helps to save energy in buildings by providing shade, evaporation and transpiration, heat insulation and wind protection. In addition, it controls heat transfer and reduces the heat load of buildings [53], thereby reducing the power consumption of buildings and effectively realizing the energy saving effect [119]. Table 5 shows the literature on energy saving effect of vertical greening.

**Table 5.** Literature on energy saving effect of vertical greening.

| Type | Counry | Plant | Season | Reduction in Energy Consumption (%) | Method |
|---|---|---|---|---|---|
| Green Wall | Hong Kong [120] | Zoysia japonica | Summer | $30$ W/m$^2$ heat flux reduction. | Experiment |
| | Wuhan [121] | – | | $2.5$ W/m$^2$ heat flux reduction $12\%$ cooling load reduction. | |
| | Lonigo, Venice, & Pisa, Italy [22] | Shrubs, Herbaceous and Climber | | $1.5$ W/m$^2$–$70$ W/m$^2$ heat flux reduction at night. | |
| | Genoa, Italy [122] | Cistus Jessamine beauty and Cistus crispus | | $26.50\%$ | |
| | Puigverd de Lleida, Spain [123] | Rosmarinus officinalis and Helichrysum thianschanicum | Winter | $2.96$–$4.2\%$ | |

*4.3. Social Functions*

4.3.1. Cultivate Interest

Through its soft and natural characteristics, vertical greening technology eliminates the cold and hard appearance of steel and cement in the urban environment and offsets disharmonious factors such as fast-paced and high pressure in the city. It also promotes people's optimistic and comfortable form, cultivates people's interest and promotes physical and mental health [23,36,37].

4.3.2. Beautifying the City

Vertical greening has good ornamental value. It extends the urban landscape from the plane to the three-dimensional, increases the landscape level and effect, and improves the greening coverage. It also weakens the rigid shape of the building, improves the city's image, and makes its urban space more diverse and friendly [23].

## 5. Knowledge Graph for Vertical Greening System

The knowledge panorama, knowledge dynamics, and knowledge evolution of the vertical greening field are constructed using the literature measurement method. The knowledge wedge diagram of the vertical greening system is integrated, as shown in Figure 12.

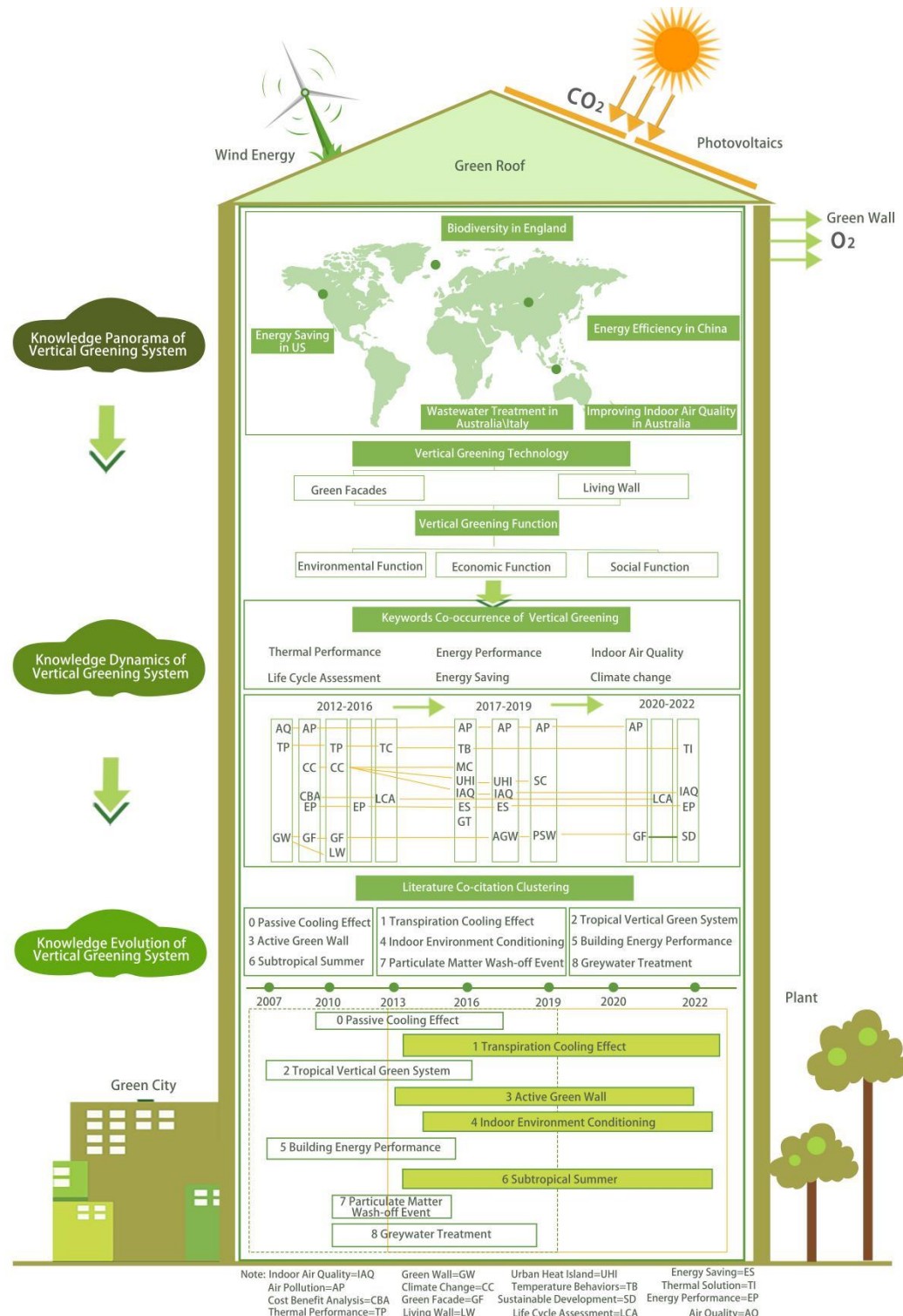

**Figure 12.** Knowledge wedge diagram of vertical greening system.

## 6. Conclusions and Perspectives

This study comprehensively considered and analyzed the vertical greening system, technology and function, combining CiteSpace, and the following results were obtained.

(1) Technology and benefits: Through the analysis of existing vertical greening technologies [124], the advantages and disadvantages of different vertical greening technologies are summarized. Secondly, sewage treatment, air quality, cooling and cooling; economic

benefits and remarkable energy-saving effect [125–127]; and social benefits have been instrumental in fostering sentiment and beautifying the city.

(2) Research hotspots and development history: This paper uses the keyword co-occurrence analysis function to identify the keywords of the vertical greening knowledge base, among which "Urban heat island", "Thermal performance", "Energy performance", "Climate change", "Assessment tools", and other keywords are research hotspots. The evolution of research hotspots has mainly gone through three stages: the initial stage, the rising development and the durable development. Obviously, air quality [128], microclimate regulation and energy issues have traditionally been the focus of attention. At the same time, more attention should be paid to the comfort of the user group.

(3) Research directions in different countries: Due to differences in countries, regions and climates, research priorities are different. In China, it is mainly the energy-saving benefit and evaluation of vertical greening. In Australia and Italy, the living wall system will be used for waste treatment and indoor space application.

(4) Research progress and future directions: The time characteristics of literature clustering and co-cited relationship and the average time of various literature can be concluded that the current research fields of the vertical greening systems are cooling, active system and indoor space.

The unique value of this paper lies in the systematic analysis of the vertical greening field, and the understanding of the current vertical greening technology and benefits, and the use of the quantitative analysis function of CiteSpace to build a knowledge map of vertical greening construction based on keywords, clustering, countries and timelines. To further enhance the vertical greening knowledge map provided by this study, we can conduct pertinent research and regularly update the data.

In future research, green energy (solar energy, wind energy, water energy) will be used to make the vertical greening system form an integrated system construction system of natural energy and natural resources to reduce the consumption of natural resources and energy. In addition, develop more intelligent vertical greening technology to facilitate maintenance [129] and play a greater role, so as to achieve sustainable urban ecological development.

**Author Contributions:** Conceptualization, P.W., Y.H.W. and W.T.C.; methodology, P.W.; validation, Y.H.W., C.Y.T. and W.T.C.; formal analysis, P.W. and S.L.; investigation, P.W.; resources, P.W., Y.H.W., C.Y.T. and W.T.C.; data curation, P.W. and S.L.; writing—original draft preparation, P.W.; writing—review and editing, P.W., Y.H.W., C.Y.T. and W.T.C.; visualization, Y.H.W., C.Y.T. and W.T.C.; supervision, Y.H.W., C.Y.T. and W.T.C.; project administration, Y.H.W.; funding acquisition, Y.H.W. All authors have read and agreed to the published version of the manuscript.

**Funding:** This work acknowledges financial support by Universiti Malaya (UM) via Faculty Research Grant (Grant No. GPF060B-2020).

**Institutional Review Board Statement:** Not applicable.

**Informed Consent Statement:** Not applicable.

**Data Availability Statement:** Not applicable.

**Conflicts of Interest:** The authors declare no conflict of interest.

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
