# Peer review of "Vertical Greening Systems: Technological Benefits, Progresses and Prospects"

_sustainability, doi:10.3390/su142012997_

Round 1
Reviewer 1 Report
The authors have submitted a paper titled “Vertical Greening System: Technological Benefits, Progresses, and Prospects”. In this paper, the relevant literature from 2012 to 2022 has investigated the development status and future trends of vertical greening technology. The paper is generally well written and structured in shape. This paper has the potential to be accepted, but some important points must be clarified or fixed.
· The abstract part needs a better write-up. Please provide the overall summary of the written manuscript. Adjust L-25, “In order to promote vertical greening technology.”
· The relevant and up-to-date review of the literature is missing in the introduction section. Try to provide better citations that will help place the current work in a better context of scientific progress.
· Try to relate the suitability of your work by discussing a few state-of-the-art papers from Sustainability in this field.
· Figure 4. Data flow chart for?
· The fitting line in Figure 5 with R2 of 0.43 is a misleading fact. Please remove the fitting line and discuss it accordingly.
· "Figure 6. keyword co-occurrence" and "Figure 9. Literature co-citation clustering and timeline." Try to caption in a more meaningful way. Usually, figures, tables, abstracts, and conclusions should be standalone readable.
· Figure 7. Keyword TimeZone View" The figure is not clear. Explain the research issue you want to discuss with this figure. improve the graphics and make it readable.
· Figure 8. shows the map of transnational cooperation. Try to improve the caption of the figure once again. Please refer to a few high-quality scientific articles.
The findings and outcome of the paper are missing. please give sharp findings and outcomes of this paper in the abstract and conclusion section.
· It is suggested to adjust these headings: (1) Climbing plants (2) Living walls and (3) Hydroponic systems into a main respective part (2.2.1. Plants).
· L-276, please replace the word “invoked” with “involved”.
· L-278, please adjust the line, “Data collection……2022.”
· Please mention the number of research and review articles obtained from the web of science.
· Figure 1, Please replace the “Outdoor/Indoodr Space” with the “Outdoor/Indoor Space”.
· Figure 4, Please use a different color scheme for a better understanding of the flowchart. Revise the caption of the figure.
· Please replace “the core datacase” with “the core database” in Figure 5.
· L-305, please adjust Fig.6 in the text.
· Please adjust the text size in Figure 6 for better visualization. Adjust the caption of the figure.
· Figure 7 could be better improved with graphical improvements in each of its parts. Adjust the caption of the figure.
· Figure 8, Please revise the caption of the figure in a more sophisticated way.
· Figure 9, Please elaborate more on this figure and try to merge it.
· Figure 10, Caption of the figure is missing. Please revise the figure with a better color scheme. Make appropriate parts of the figure as well.
· Figure 11, Please be more cautious about the use of capitalization in captions. It will be good to discuss the figure in detail in the text portion.
· Table 2, Please revise the text size in the table and try to adjust the page size.
· The overall writing is not so good. The manuscript lacks major grammatical and spelling mistakes in certain portions of the text. The conclusion section could be concise with a better technical writeup and layout.
Author Response
Please find the response letter in the attachment.

Reviewer 2 Report
The manuscript merits publication to Sustainability if the authors take into consideration some remarks:
- For better visualization, increase the size of figures (eg: 3, 6, 7, 9, 12) / tables (eg: 2) or modifying the colors.
The article contains enough information to be useful, published and of interest to an international reader.
The article understood and identified the fundamental literature of the investigated scientific area.
The article is well based on the basic theory, the research is well designed and the methods are appropriate.
The results are presented clearly and correctly. The conclusions reflect the issues addressed. Tables and figures deepen and clarify the text.
The article identifies the implications of the investigation. The conclusions can be used in a simple and practical way.
The article is clear and uses terminology from the scientific area under study.
Author Response

(The authors gave the same response as above.)

Round 2
Reviewer 1 Report
The paper has been revised accordingly.